# A Hybrid Power-Rate Management Strategy in Distributed Congestion Control for 5G-NR-V2X Sidelink Communications

**DOI:** 10.3390/s23156657

**Published:** 2023-07-25

**Authors:** Jiawei Tian, SangHoon An, Azharul Islam, KyungHi Chang

**Affiliations:** Department of Electrical and Computer Engineering, Inha University, Incheon 22212, Republic of Korea

**Keywords:** 5G-NR-V2X, distributed congestion control, power-rate management, vehicle-to-vehicle communication

## Abstract

The accelerated growth of 5G technology has facilitated substantial progress in the realm of vehicle-to-everything (V2X) communications. Consequently, achieving optimal network performance and addressing congestion-related challenges have become paramount. This research proposes a unique hybrid power and rate control management strategy for distributed congestion control (HPR-DCC) focusing on 5G-NR-V2X sidelink communications. The primary objective of this strategy is to enhance network performance while simultaneously preventing congestion. By implementing the HPR-DCC strategy, a more fine-grained and adaptive control over the transmit power and transmission rate can be achieved. This enables efficient control by dynamically adjusting transmission parameters based on the network conditions. This study outlines the system model and methodology used to develop the HPR-DCC algorithm and investigates its characteristics of stability and convergence. Simulation results indicate that the proposed method effectively controls the maximum CBR value at 64% during high congestion scenarios, which leads to a 6% performance improvement over the conventional DCC approach. Furthermore, this approach enhances the signal reception range by 20 m, while maintaining the 90% packet reception ratio (PRR). The proposed HPR-DCC contributes to optimizing the quality and reliability of 5G-NR-V2X sidelink communication and holds great promise for advancing V2X applications in intelligent transportation systems.

## 1. Introduction

As connected vehicles become increasingly prevalent, managing traffic flows efficiently is crucial to preventing congestion. Intelligent transportation systems (ITS) and V2X communications play vital roles in achieving effective, secure, and sustainable transportation. 5G-NR-V2X sidelink communications have emerged as a notable solution for facilitating direct and dependable communication between proximate vehicles, without the need for a base station or cellular network. Despite its advantages, supervising communication within 5G-NR-V2X sidelink networks presents challenges due to the dynamic nature of vehicular traffic and the potential for congestion. Various congestion control methods have been proposed to tackle these challenges, including DCC algorithms, which aim to optimize network resource utilization while maintaining fairness and stability [1]. DCC, a cross-layer mechanism standardized by the European Telecommunications Standards Institute (ETSI), is extensively utilized in the ITS sector [2]. However, there remains potential for enhancing its performance in terms of improved network utilization, minimized packet loss, and equitable bandwidth allocation. This paper focuses on investigating and developing an optimized DCC algorithm for 5G-NR-V2X sidelink communications within the context of ITS. The main objective is to augment the performance of DCC algorithms by integrating transmission power control (TPC) and transmission rate control (TRC) methods into a hybrid control framework [3]. By combining the strengths of both TPC and TRC, this approach aims to facilitate superior network utilization and congestion management in vehicular networks, leading to more efficient and sustainable transportation systems.

Within the ITS context, the exchange of cooperative awareness messages (CAMs) and event-triggered decentralized environmental notification messages (DENMs) plays a vital role [4]. Periodically broadcast CAMs convey essential information such as vehicle position, speed, acceleration, and other relevant data, which are crucial for safety and traffic efficiency applications. To address hazardous road situations promptly, DENMs are also required. Both CAMs and DENMs are transmitted on the control channel (CCH), dedicated to cooperative road safety [5]. However, the use of a shared control channel can lead to radio congestion in scenarios with high vehicle density. To address this issue, the DCC framework is introduced as a solution for alleviating control channel congestion. The DCC framework achieves this by regulating the message rate, transmission power, and data rate of periodic messages. Operating as a cross-layer mechanism, the DCC encompasses congestion control at both the network and MAC layers, utilizing information from the physical and network layers to manage congestion effectively. The protocol employs a sliding window mechanism, dynamically adjusting the window size based on the network congestion level. The DCC protocol features two distinct approaches for managing congestion control: adaptive and reactive [3]. The DCC Adaptive approach, prescribed by the ETSI, adaptively adjusts DCC parameters based on real-time evaluations of network conditions [5]. By assessing the current network congestion status using gathered metrics, this method modifies communication parameters according to the evaluation results. In contrast, the DCC Reactive approach is designed to respond to specific network events or triggers as shown in Figure 1. Instead of proactively monitoring and adjusting communication parameters based on real-time network conditions like the DCC Adaptive approach, the DCC Reactive approach operates through a set of predefined rules or policies. Upon detecting a triggering event, this reactive method promptly takes action to mitigate congestion by adjusting communication parameters, such as transmission power, packet size, or transmission intervals. To achieve more reasonable resource allocation, it is also possible to refine each state parameter by adding multiple Active states [6].

Upon analyzing existing congestion control schemes, this research introduces a hybrid congestion control scheme named the HPR-DCC for V2X communications by integrating the DCC framework and rate-power-based control mechanism. The hybrid approach leverages the complementary strengths of TPC and TRC, mitigating the limitations of each technique. TPC helps to maintain reliable communication by adjusting the transmission power based on channel conditions, while TRC regulates the data transmission rate to prevent congestion and ensure fair resource allocation. By combining the advantages of the TPC with the TRC, the HPR-DCC scheme offers enhanced performance in terms of congestion mitigation, reliable communication, and fair resource allocation. Furthermore, the HPR-DCC enhances adaptability to dynamic network conditions by dynamically switching between TPC and TRC based on real-time feedback and network metrics. Thus, the TPC and TRC hybrid control DCC algorithm provides a robust and efficient solution for congestion control in wireless networks, addressing the challenges of varying channel conditions and network congestion while optimizing resource utilization. The primary contributions of this paper are twofold: Proposing a novel method that combines the strengths of both TPC and TRC to achieve efficient and reliable communication;Evaluating the performance of the proposed scheme through simulations, comparing it to existing congestion control schemes and demonstrating its effectiveness in alleviating congestion across various traffic situations.

This paper offers valuable insights into designing efficient and reliable congestion control schemes for 5G-NR-V2X, which are essential for implementing future intelligent transportation systems. And the rest is organized into following sections: Section 2, provides an overview of congestion control in 5G-NR-V2X sidelink communications, explores transmission power and rate control schemes in wireless networks, and introduces the HPR-DCC algorithm. Section 3 presents the system model and assumptions, elaborates on the proposed algorithm’s design, and analyzes the control scheme’s stability. Section 4 outlines the simulation setup and scenarios, defines performance metrics and evaluation criteria, offers experimental results and analysis, and compares the proposed scheme to alternative congestion control strategies. Finally, it recapitulates the principal contributions and outcomes, discusses practical implications and applications, and recommends directions for further research.

## 2. Related Works

In recent years, significant attention has been given to the development and implementation of DCC methods in vehicular ad hoc networks (VANETs). The primary goals are to manage network congestion and enhance the reliability of V2X communications [5,7]. Comparing various DCC methods is challenging due to the diverse control strategies and performance metrics in the literature. To tackle this issue, standardization organizations like ETSI have developed performance evaluation standards and proposed a unified cross-layer DCC framework. This research will focus on DCC methods that optimize two primary metrics: CBR [8] and packet reception ratio (PRR) [9]. This study aims to provide a comprehensive understanding of the current state-of-the-art in DCC for VANETs and identify potential avenues for further research and development. A lower CBR value indicates a more efficient utilization of the communication channel, contributing to reduced network congestion. Conversely, PRR is calculated by dividing the number of received data packets by the number of sent data packets [9], thereby serving as a measure of successful communication within the network. A higher PRR value signifies better signal reception, ensuring that critical safety and traffic information is effectively communicated among vehicles. Our study pays particular attention to these optimization objectives and the resulting control strategies. Through the investigation of these approaches and their impact on network efficiency, our objective is to offer an in-depth insight into the contemporary advancements in DCC for VANETs, while pinpointing possible directions for future exploration and progress.

The authors of [10] present the linear message rate integrated control (LIMERIC) algorithm, an effective solution that addresses fairness concerns and shows adaptability in various complex scenarios. Building upon the LIMERIC algorithm, the error model based adaptive rate control (EMBARC) algorithm is introduced in [11], which improves the LIMERIC approach by dynamically adjusting the transmission rate according to vehicular movement. In [12], the researchers suggest integrating beaconing into the vehicular networks framework and modifying the beacon frequency as well as the transmission rate to efficiently manage traffic congestion.

To ensure that DCC is applicable to a broader array of scenarios, including those with a heightened focus on security, a substantial quantity of broadcast beacons is necessary. Consequently, the TRC mechanisms might prove insufficient in fulfilling all aspects of operational security requirements. Torrent-Moreno and colleagues have shown that effective TPC is crucial for optimizing channel utilization while mitigating security concerns arising from channel saturation. To address this issue, they propose a solution called distributed fair power adjustment for vehicular environments (D-FPAV) [13]. This control scheme adheres to stringent fairness principles, ensuring prioritized transmission for high-priority data and equitable transmission conditions for other vehicles based on the prevailing channel conditions. The design of TPC is characteristically complex, as it encompasses rapidly evolving networks. Nevertheless, it is particularly well-adapted for streamlined and linear network topologies, such as those found in platooning scenarios. In [14], the authors explore various communication strategies for platooning by employing synchronized communication slots in conjunction with TPC techniques. They subsequently compare their proposed method with alternative beaconing solutions, including static beaconing and conventional ETCI DCC for automated platooning applications. The simulation results indicate that the suggested approach can effectively reduce collisions. Moreover, the researchers examined a mixed scenario wherein some vehicles simultaneously accessed the channel using ETSI DCC. They found that the performance of their proposed solution remained unaltered, while the ETSI DCC performance was significantly impacted. In [15], the authors present a DCC algorithm that integrates a priority model and adjusts the beacon transmission rate. This algorithm effectively manages congestion’s influence on vehicle safety by guaranteeing the reliable and timely reception of safety information. In [16], the authors present a novel approach to enhance the object filtering process of collective perception by considering DCC awareness. This approach dynamically adjusts the message size based on DCC constraints and, consequently, the message generation rate. The comparison with the existing ETSI design shows that this design improves the perceived quality and reduces the message generation rate. In [17], the authors introduce a traffic density-based congestion control algorithm (TDCCA) that incorporates vehicle IDs into their respective CAMs and utilizes TRC-based DCC to enhance model parameter efficiency. The algorithm considers a range of network conditions, from non-saturated to saturated, as well as sparsely dispersed and congested networks. The proposed approach demonstrates improved performance in terms of PRR and latency.

While the majority of conventional DCC algorithms rely on CBR as the criterion for determining control parameters, a single-measurement approach is insufficient due to the myriad factors influencing channel load, which consequently leads to issues such as fairness. In [18], the authors introduce Rate-OPT and Power-OPT algorithms for dedicated short-range communication (DSRC), demonstrating that their coordinated and alternating application results in enhanced channel utilization and packet transmission rates. This integrated congestion control algorithm optimizes channel load usage by dynamically allocating transmission range and rate in response to vehicle density. In [19], the authors introduce a combined transmission power and rate control strategy, which deprioritizes the use of TPC as the primary response mechanism. In contrast to a purely TPC-based approach, this strategy lessens the reliance on precise transmission power adjustments and demonstrates that it can efficiently approximate the optimal control parameter configuration for load adaptation across individual channels. A joint power and rate algorithm in [20], which can set different priorities for different vehicles, introduces fairness into V2V communication, and conducts simulation verification through multiple scenarios. In [21], the authors propose a perception-based hybrid beacon algorithm that utilizes the driver’s state as a reference condition and broadcasts it to nearby vehicles, then makes joint decisions to adjust the transmission range and power in order to enhance safety. In [22], the authors propose an approach that integrates real-time traffic flow sensing with channel congestion status, utilizing distributed network utility maximization to improve channel utilization. In [23], the authors introduce the POSACC algorithm, prioritizing location accuracy and communication reliability as paramount metrics. They effectively manage the beacon rate and transmission power, resulting in enhanced efficiency.

The HPR-DCC integrates the merits of both TRC and TPC, facilitating improved congestion control and maintaining CBR within the convergence range. TPC allows for power-efficient communication by reducing transmit power when the channel conditions permit, while TRC optimizes the transmission rate to maximize throughput when the channel quality is favorable. This system offers an adaptable balance between energy consumption and network efficiency. However, the most optimal coordination and equilibrium between the TPC and TRC depend on the specific system requisites and the characteristics of the adaptive scenario in which they will operate. Consequently, the full potential of the HPR-DCC is realized through careful consideration and customization of the system according to the specific needs and conditions of the intended application. While it holds promising benefits, it is important to note that the coordination between TPC and TRC can be a complex process. The HPR-DCC’s potential limitations thus lie in this inherent complexity of achieving the optimal balance between power and throughput efficiency. Several factors affect the quality of channel communication, such as vehicle speed, vehicle density, and signal transmission distance. Relying solely on CBR as the foundation for the control algorithm proves inadequate for ensuring equitable communication among all users. This study introduces the incorporation of additional parameters alongside CBR to holistically assess state transitions, guaranteeing a more equitable allocation of resources for vehicles experiencing identical states.

## 3. System Design and the Proposed HPR-DCC Methodology

### 3.1. System Model and Assumptions

The proposed system model aims to estimate vehicular parameters by emphasizing the adjustment of transmission power and transmission rate. This is achieved through the assimilation of received messages and distance information from neighboring vehicles, as illustrated in Figure 2. This model incorporates critical parameters, such as neighboring vehicles’ transmission power, distance of received messages, and estimated path loss (PL), which are instrumental in determining the optimal transmission power necessary for achieving the desired level of awareness within the target vehicle’s awareness range. The system model operates on several fundamental assumptions. Primarily, it assumes that the transmission power is modulated based on the estimated PL value, with the goal of achieving the target awareness percentage within the awareness range, while not considering the impact of frame error rate. Subsequently, the model presumes that the PL value estimation relies on messages from a sufficient number of neighboring vehicles, enabling target-aware transmission for vehicles that have not received any messages. Lastly, under extreme circumstances where the distance between vehicles is minimal and path loss is substantial, the transmission power will be maintained at a prominent level to ensure the preservation of the target awareness range. The control mechanism embedded within the proposed system model incorporates the calculation of CBR, received messages, and distance information, as well as the computation of transmission power necessary to attain the target awareness percentage. Moreover, the estimation of the PL value is employed to adjust the transmission power for target-aware transmission within the awareness range. This ensures that the vehicle maintains awareness of the target vehicle throughout the communication process.

### 3.2. HPR-DCC Design and Strategies

In the design of the HPR-DCC algorithm, a three-fold approach is employed to optimize performance and efficiency. Firstly, power adaptation for awareness control is facilitated through the component, which dynamically adjusts the transmission power in accordance with the target awareness range dictated by the application context. By estimating the PL, the algorithm is able to modulate transmission power to satisfy awareness prerequisites, even for vehicles in the ‘worst’ channels that have not exchanged messages. Secondly, the HPR-DCC incorporates rate control by leveraging the state machine. Owing to its ability to converge towards fair and efficient channel utilization, the state machine regulates the forthcoming message rate to sustain the CBR beneath the predetermined threshold. Lastly, the HPR-DCC combines both power and rate control, adjusting the subsequent transmission power on the basis of the current path loss for each message obtained from neighboring vehicles. This process considers the target awareness percentage when determining the appropriate transmission power level. Simultaneously, the algorithm adapts the rate by considering the existing message rate and channel load, represented by the CBR. This comprehensive approach allows for enhanced adaptability and performance in the context of the HPR-DCC algorithm design and features a simple structure.

To adjust transmission power in response to congestion control and vehicle demand, the proposed algorithm makes a joint decision by calculating PL and current state CBR for state switching. The specific explanation is as follows:

Assume that vehicles transmit power at time *t*: $P_{i}^{Tx}\left(t\right)$; target awareness range of vehicle: $TR_{e}\left(t\right)$; target awareness percentage of vehicle: $TA_{e}\left(t\right)$; and shadowing coefficient: S. For each received message, calculate $d_{ij}\left(t\right)$, the distance between vehicle and *i*th neighbor at time *t* when received the message *j*. Compute $PLE_{ij}\left(t\right)$, the PL for message *j* from neighbor, by Equation (1): (1)$$PLE_{ij}\left(t\right)=\frac{PL\left(t\right)}{10{\;\log}_{10}\left(\frac{4\Pi }{\lambda }d_{ij}\left(t\right)\right)},$$
where λ is the signal wavelength and $PL\left(t\right)$ is calculated by Equation (2):(2)$$PL\left(t\right)=P_{i}^{Tx}\left(t\right)-P_{j}^{Rx}\left(t\right),\;$$
where $P_{i}^{Tx}\left(t\right)$ represents the transmit power of neighbor *i* and $P_{ij}^{Rx}\left(t\right)$ denotes the receive power of *j*th message form neighbor *i*.

Then, calculate the received power required as Equation (3): (3)$$P_{r}\left(t\right)=\frac{P_{t}\cdot G_{t}\cdot G_{r}\cdot S}{PL\left(t\right)},\;$$
where $P_{t}$ is the transmitted power, $G_{t}$ is transmitter antenna gain, and $G_{r}$ is the receiver antenna gain. Then, set the transmission power for next time $\left(t+1\right)$:(4)$$P_{sorted}{}_{e}^{Tx}=sort_{\forall i,j\in N}\left(P_{r}\left(t+1\right)\right),\;$$
where $P_{r}\left(t+1\right)$ is calculated as:(5)$$P_{r}\left(t+1\right)=P_{sorted}{}_{e}^{Tx}\left[round\left(TA_{e}*N\right)\right].\;$$

In Equations (4) and (5), sort the necessary transmission power to each neighboring node and select the appropriate power level for transmission.

In the proposed algorithm, state switching is jointly determined by PL and CBR, this joint decision-making facilitates ensuring fairness in policies. Multiple states are established, with each active state allocating the appropriate transmission power and transmission rate according to the current channel conditions [24]. The detailed state transitions are presented in Table 1.

### 3.3. Proposed HPR-DCC Algorithm

Algorithm 1 outlines the steps of the proposed HPR-DCC, a hybrid power and rate control DCC algorithm that can dynamically adjust the transmission power and rate of vehicular communication systems.

The transmission power control component of the proposed HPR-DCC assigns appropriate transmit power to neighboring vehicles that were already connected to the node in the previous time step. This assignment is based on the current path loss and path loss exponent. If vehicles are not neighbors in the previous time step, a default value is used as the transmission power. The HPR-DCC then sorts these power values in ascending order and selects the smallest value that meets the target awareness percentage as the next transmission power. In terms of rate control, the HPR-DCC adjusts the communication rate according to the current channel load (CBR), which is calculated as the ratio of received messages to the channel capacity. The state machine rate control mechanism can also adjust to diverse congestion scenarios and assign suitable transmission rates to vehicles. To maintain efficiency, the power and rate control decisions collaboratively “share the load” under high CBR conditions. The balancing of this relationship between the target and current beacon rate and awareness is determined by the coefficient *γ*, which is currently set to 1. That means the Tx power and Tx rate share the equal weight. The coefficient *γ* will be utilized to effectively coordinate the control module in this context. When the detected transmission power error rate $\delta_{P}$ surpasses the transmission rate with the coefficient $\gamma \delta_{R}$, a higher level of transmission power will be employed as a means to alleviate it. Conversely, if the detected transmission power error rate with coefficient $\gamma \delta_{R}$ is below the transmission rate, the current transmission power level will be maintained without any adjustment.
**Algorithm 1:** Hybrid Power and Rate Control DCC algorithm.**Neighbor transmission power detection**1: **Calculate:**
PLt=PiTxt−PjRxt
2:      PLEijt=PLt10 log104Πλdijt
3: **if** Neighbore→it ∈Neighboret−1 **then**4:  Prt=Pt·Gt·Gr·SPLt
5: **else**6:  PsortedeTx=sort∀i,j∈NPrt+1
7:  Prt+1=PsortedeTxroundTAe∗N
**Congestion detection process**8:  **Sensed busy if** ∑Pri>Sbusy **then**9:  Record Tbusy
10:   **Calculate CBR every** TCBR=100 ms
11:   CBR = Tbusy/TCBR
**Transmission power allocation**12:   **if** CBRt<CBRTh **then**13:    Apply Prt+1
14:   **else**15:    **if** Prt+1≤Prt **then**16:      Apply Prt+1**Transmission rate allocation**17:    **if New Tx power apply then**18:      State switching Active19:    **else**20:      Keep default Tx rate**Coordinated Control Module**21:   **Calculate error rate of Tx Power and Tx Rate**22:   δP=PRt−PERt23:   δR=TRt−BRtTRt24:   **if** δP≥ γδR **then**25:      Apply Prt+126:   **if** δP<γδR **then**27:      Keep Tx power the same

Additionally, the HPR-DCC is designed to manage channel load by preventing significant increases in channel load caused by a sudden growth in the target awareness range. However, the proposed HPR-DCC allows safety-critical messages generated during hazardous events to be transmitted at high power and rate, by passing the standard restrictions. This ensures that crucial information is promptly communicated in emergency situations, thereby enhancing the overall safety and reliability of the vehicular communication system.

In this study, the parameter settings of the proposed HPR-DCC algorithm are calibrated to account for the multifaceted aspects of V2V communication systems, thereby ensuring an accurate representation of scenarios, with the main parameters shown in Table 2. The time step duration is set at 200 milliseconds, providing a suitable time resolution for capturing the dynamic interactions within V2V networks. Both the target range and target awareness are defined as context-dependent variables, which typically fluctuate between 20 and 500 m, and 50% to 100%, respectively, depending on the specific application context. This flexible approach allows the algorithm to adapt to various V2V communication scenarios and capture the nuances of different vehicular environments. Furthermore, the maximum transmission power is confined to a range of 0 to 23 dBm [25], adhering to the standard constraints for V2V communication radios. This restriction ensures that the proposed algorithm operates within the acceptable power limits established for vehicular communication systems, mitigating potential interference or signal degradation issues. And, the maximum beacon rate (BR) is specified within a range of 1 to 10 Hz, representing a typical range for cooperative messages in V2V communication systems. By constraining the beacon rate within this range, the algorithm effectively accommodates the requirements of V2V communication, facilitating efficient and reliable data exchange between vehicles. Then, set the subcarrier spacing as 15 kHz for providing better time domain resolution, reduce the influence of multipath fading on the signal, and improve the anti-interference ability of the system.

### 3.4. Feasibility Analysis of the Proposed HPR-DCC Algorithm

The feasibility of using a hybrid control strategy that merges TPC and TRC for con-gestion management is supported by multiple factors. First, TPC provides precise control over wireless nodes’ transmission power, enabling adjustments in communication range and link quality. This capability effectively mitigates congestion by reducing interference and contention within the network. Additionally, TPC ensures efficient power allocation, optimizing energy consumption and extending the network’s operational lifetime. Second, TRC allows for regulation of data transmission rate, enabling dynamic channel capacity control and ensuring optimal network resource utilization. By adopting a hybrid control strategy, the limitations of individual control methods, such as exclusive reliance on power or rate control, are overcome. Moreover, TRC offers flexibility in managing traffic, as it can promptly adapt to changing network dynamics. By combining TPC and TRC in a hybrid control strategy, the limitations of individual control methods are effectively addressed. The hybrid approach uses the precision of power control and the adaptability of rate control, resulting in a more versatile and resilient congestion management solution. The HPR-DCC is expected to improve network throughput, decrease packet loss, and minimize delays, making it a promising solution for congestion management in wireless networks.

## 4. Performance Evaluation and Discussion

### 4.1. Evaluation Parameters

To manage channel congestion, the 3GPP standard defines a metric named CBR, as well as potential mechanisms for leveraging these metrics to mitigate channel congestion [20]. The CBR is a measure of the portion of time the channel is busy transmitting data. CBR is useful for quantifying the level of channel congestion and can be utilized to implement congestion control mechanisms. By monitoring the CBR, the network can dynamically adjust the allocation of channel resources and regulate the transmission of data to avoid congestion. The CBR is calculated every $T_{CBR}$ = 100 ms as follows: (6)$$\mathrm{CBR}=T_{busy}/T_{CBR},$$

In Equation (6), the channel occupancy of $T_{busy}$ is dynamically updated at the beginning or end of each transmission for every vehicle. The calculation of $T_{busy}\;$ is determined based on whether the channel is sensed as busy or not. Specifically, the channel is considered busy if the received power level $P_{r_{i}}$ is greater than the sensitivity threshold $S_{busy}$. The sensitivity threshold $S_{busy}$ is set to −94 dBm [26].

The degradation of PRR is a common issue in vehicular networks as vehicle density increases. Higher PRR guarantees more reliable communication and is calculated as below Equation (7):(7)$$\mathrm{PRR}=\frac{P_{r}}{P_{r}+P_{SL}+P_{TL}},$$
where $P_{r}$, $P_{SL}$ and $P_{TL}$ represent the total received packets, SINR packet loss, and packet loss of transmitting, respectively.

The degradation of PRR is a common issue in vehicular networks as vehicle density increases. DCC algorithms are commonly used to mitigate PRR reduction, which typically involves reducing the CBR. However, lowering the CBR could result in reduced throughput performance. Hence, the aim of this study is to determine the optimal packet transmission power and packet transmission rate that can maximize the aggregate PRR of vehicle user equipment (VUE), while also maintaining the CBR to a predetermined target, even in high-vehicle-density scenarios, through the HPR-DCC algorithm. 

### 4.2. Simulation Setup and Scenarios

Consider a C-V2X network consisting of a number of VUE that are spatially distributed using a 1-D Poisson point process with a variable density. The highway length is set to 2 km with three lanes in each direction, and the width of each lane is 4 m [25,26,27]. In this scenario, VUE moves with a predefined speed and when they reach the end of the road, they loop around and enter the opposite direction, as shown in Figure 3.

The VUE periodically broadcasts CAMs via V2V sidelink communication. For the C-V2X system, single carrier frequency-division multiple access (SC-FDMA) is used in a 10-MHz-wide channel. Each VUE automatically selects radio resources using the allocation procedure of SB-SPS. The reselection counter is randomly selected to be between 5 and 15 in the SB-SPS [27,28,29,30]. And the self-interference cancellation coefficient is set as −110 dB to effectively eliminate interference between its own transmission and reception. The average duration of the interval for the CBR calculation is set to 100 ms to ensure the provide more real-time performance metrics of channels. To evaluate the efficacy of HPR-DCC under varying traffic congestion conditions, we manipulate vehicle densities to 40, 80, and 120 vehicles/km while maintaining an average speed of 140 km/h. This manipulation represents low, medium, and high-traffic scenarios, respectively. We further set the standard deviation of vehicle speeds at 3 km/h to approximate real-world traffic conditions. In the initialization of the simulation, under low-stress conditions, we establish the initial transmission power at 10 dBm and the initial transmission rate at 10 Hz [31,32,33,34]. As traffic conditions transition, HPR-DCC facilitates the necessary adjustments. Consequently, the transmission power fluctuates within a 0–23 dBm range, and the packet transmission frequency range varies between 1 and 10 Hz [31]. The main simulation parameters are the same as those listed in Table 3.

This simulation scenario, by using the enhanced LTEV2Vsim simulator [35], allows us to evaluate the proposed DCC; thus, we can verify its effectiveness in maximizing packet reception rate and minimizing the CBR while still maintaining a low collision rate.

### 4.3. Experimental Results and Analysis

The curves representing the CBR performance indicate that the proposed HPR-DCC consistently surpasses both the case without DCC and the original DCC. Nevertheless, the extent of improvement, when contrasted with the original DCC scheme, fluctuates depending on the specific congestion scenarios. Figure 4a demonstrates that the maximum convergence value of the proposed DCC is 30% when vehicle density is 40 veh/km, which does not significantly enhance the original DCC’s improvement of 32%. Figure 5a, when vehicle density is 80 veh/km reveals that although the proposed HPR-DCC marginally outperforms the original DCC in congestion control, their maximum convergence values are nearly identical. In a highly congested setting with a vehicle density of 120 veh/km, as illustrated in Figure 6a, the proposed DCC attains a maximum CBR of 64%, while the original DCC scheme reaches a maximum CBR of 70%, exhibiting the most considerable gain of about 6%. In this research, we adopt a hybrid control approach that combines TPC and TRC mechanisms for congestion control and employs multiple active states to enhance channel load mitigation and vehicular communication. The proposed HPR-DCC leverages hybrid and distributed control design, offering enhanced control capabilities in high channel load scenarios. Through adaptive transmission facilitated by TPC, the transmit power is dynamically adjusted to compensate for channel quality variations, ensuring optimal signal reception even in challenging environments. As a complementary feature to TPC, TRC effectively prevents network overload and packet loss while maintaining a balance between throughput and reliability. As a result, when compared to the conventional DCC method, our proposed DCC strategy consistently produces lower CBR values throughout the simulation. 

Since PRR serves as a critical indicator of packet transmission success rate, only PRR results exceeding 90% are considered to ensure a fair comparison. In comparison to the non-DCC scheme, the HPR-DCC scheme displays enhanced PRR performance across various test environments. In Figure 4b, in low congestion scenario when PRR is equal to 90%, the distance is extended by approximately 10 m relative to the original DCC scheme. A comparable outcome is depicted in Figure 5b, suggesting that the performance improvement of the HPR-DCC is not evident in low- and medium-congestion situations. In the high-congestion test scenario, the results presented in Figure 6b disclose that the effective reception distance of the proposed DCC is extended by 20 m compared to the original DCC, thereby significantly improving the system’s signal reception performance. This enhancement arises from the incorporation of the distance between the control vehicle and neighboring vehicles into the state-switching strategy of the proposed HPR-DCC algorithm. When the gain of transmit power to the effective transmission distance surpasses a certain threshold, TRC is employed as an optimization supplement. Consequently, this improves channel conditions for vehicles located beyond the range of the original DCC.

The results demonstrate that the proposed HPR-DCC scheme offers several advantages over the original schemes. Specifically, it performs better in highly congested environments, exhibiting an improved transmission range and a higher packet reception rate. Additionally, the proposed scheme maintains low channel occupancy.

## 5. Conclusions and Future Work

This paper presents an optimized DCC scheme that utilizes a hybrid approach, combining TRC and TPC algorithms to create a more effective and robust congestion control solution. The HPR-DCC method dynamically allocates transmission power and rate according to the degree of congestion, which improves overall network performance and efficiently allocates available bandwidth. By incorporating PL and CBR metrics, the DCC scheme can make joint state-switching decisions, enhancing its flexibility and adaptability to various network scenarios and congestion situations. Achieving such flexibility entails real-time monitoring and detection of current network conditions, which allows for improved responsiveness to network demands and enhanced network performance. By introducing additional state-switching conditions, the DCC algorithm can be more adaptive, enabling it to better manage network instability and fluctuations. Simulation results substantiate that the HPR-DCC effectively controls the maximum CBR value within 64%, with a 6% enhancement compared to the original DCC approach and an extension of the effective signal reception distance by 20 m while maintaining a PRR of 90%.

In our future research, we intend to place greater emphasis on algorithmic complexity. This is because the coordination and synchronization between TPC and TRC necessitate implementation and tuning, which can potentially increase computational and processing overhead when compared to existing methods. Subsequently, we will evaluate the performance of the proposed hybrid control scheme in more intricate network environments, such as urban scenarios. Additionally, we aim to explore the possibility of incorporating machine learning techniques into the design of the hybrid control scheme, which could potentially enhance its adaptability and robustness. Furthermore, we will investigate the integration of other advanced technologies to improve the performance and security of the hybrid control scheme.

## Figures and Tables

**Figure 1 sensors-23-06657-f001:**
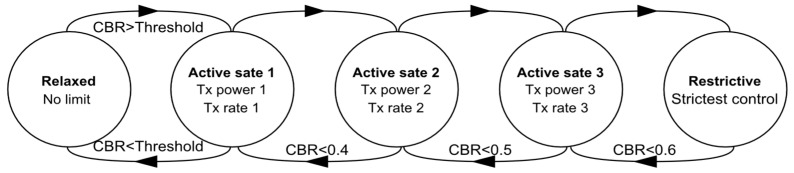
The generic outline of the reactive approach.

**Figure 2 sensors-23-06657-f002:**
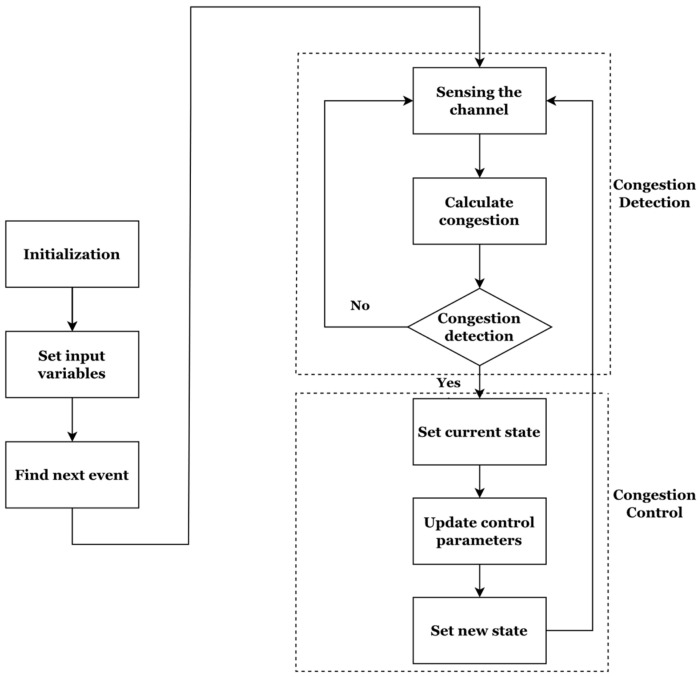
Procedure of the HPR-DCC algorithm.

**Figure 3 sensors-23-06657-f003:**
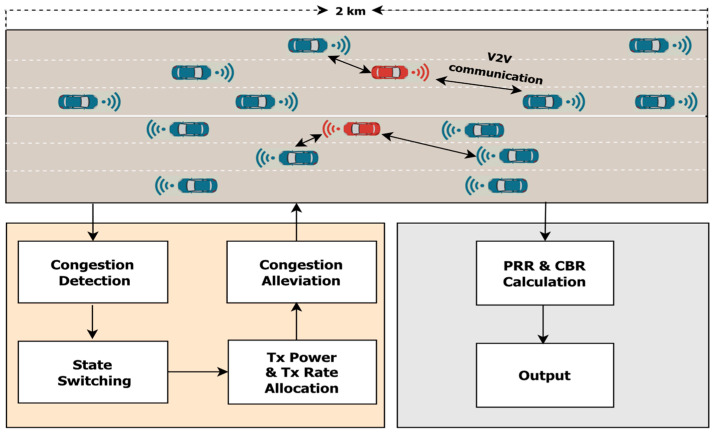
The HPR-DCC based congestion control in a highway scenario.

**Figure 4 sensors-23-06657-f004:**
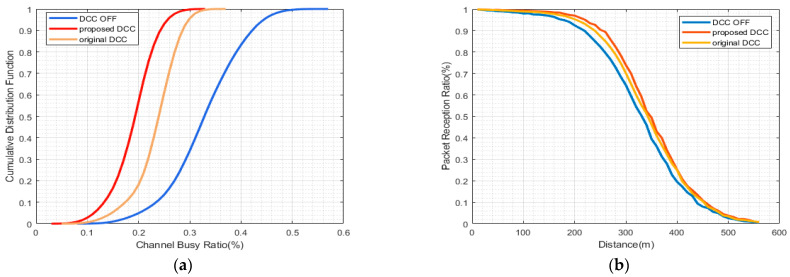
Comparison of results using HPR-DCC algorithm under vehicle density = 40: (**a**) CBR vs. CDF performance; (**b**) PRR vs. distance performance.

**Figure 5 sensors-23-06657-f005:**
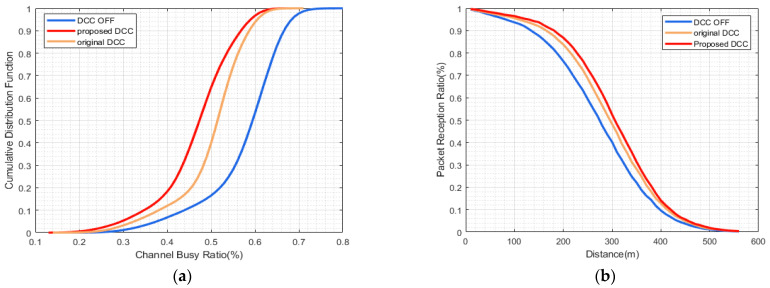
Comparison of results using HPR-DCC algorithm under vehicle density = 80: (**a**) CBR vs. CDF performance; (**b**) PRR vs. distance performance.

**Figure 6 sensors-23-06657-f006:**
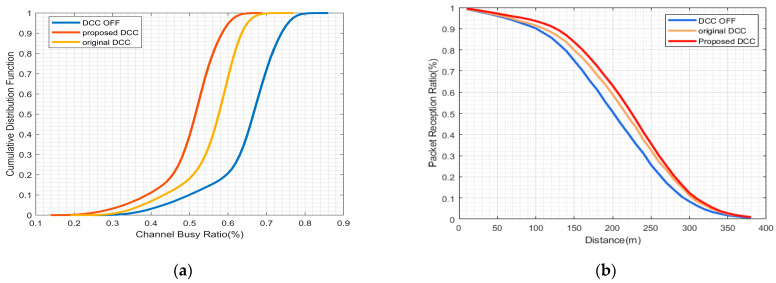
Comparison of results using HPR-DCC algorithm under vehicle density = 120: (**a**) CBR vs. CDF performance; (**b**) PRR vs. distance performance.

**Table 1 sensors-23-06657-t001:** State switching strategies of state machine.

State	CBR vs. Target	PL vs. Target	Tx Power (t + 1)
1	<	<	Apply $P_{r}\left(t+1\right)$
2	<	≥	Apply $P_{r}\left(t+1\right)$
3	<	<	Apply $P_{r}\left(t+1\right)$
4	<	≥	Apply $P_{r}\left(t+1\right)$ if ≤ $P_{r}\left(t\right)$
5	>	<	Apply $P_{r}\left(t+1\right)$ if ≤ $P_{r}\left(t+1\right)$
6	>	≥	Apply $P_{r}\left(t+1\right)$ if ≤ $P_{r}\left(t\right)$
7	>	<	Apply $P_{r}\left(t+1\right)$ if ≤ $P_{r}\left(t\right)$
8	>	≥	Apply $P_{r}\left(t+1\right)$ if ≤ $P_{r}\left(t\right)$

**Table 2 sensors-23-06657-t002:** System specifications for HPR-DCC algorithm.

Parameter	Value
Carrier frequency	5.9 GHz
Bandwidth	10 MHz
Number of subchannels	5 subchannels each 10 RBs
Subcarrier spacing	15 kHz
Subchannel size	10 RB
MCS	11
Channel model	Winner + B1
Measurement period	100 ms
Resource allocation period	100 ms
Transmitter antenna gain	3 dB
Receiver antenna gain	3 dB
Noise figure	9 dB

**Table 3 sensors-23-06657-t003:** System-level simulation parameters.

Parameter	Value
Road length	2 km
Lanes of the road	3 lanes in each direction
Average vehicle density	40, 80, 120 veh/km
Mean speed of vehicles	140 km/h
Standard deviation of speed	3 km/h
Message size	1000 Bytes
Packet generation rate	50 packets/s
Carrier sense threshold	−94 dBm
Data rate (default)	6 Mbps
Transmission power range	0–23 dBm
Packet transmission frequency range	1–10 Hz
Target awareness ratio	85%

## Data Availability

Not applicable.

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
