# Peer review of "A Hybrid Power-Rate Management Strategy in Distributed Congestion Control for 5G-NR-V2X Sidelink Communications"

_sensors, 2023, doi:10.3390/s23156657_

Round 1

Reviewer 1 Report

This paper presents an hybrid power and rate control management strategy for distributed congestion control (HPR-DCC) focusing on congestion control in 5G-NR-V2X sidelink communications.

It enhances the quality of service and reliability of 5G-NR-V2X sidelink communications, paving the way for innovative applications such as vehicle-to-vehicle (V2V) communications.

It also improves the network performance while simultaneously preventing congestion by controlling both TPC and TRC.

The authors have conducted simulations to evaluate the performance of the proposed HPR-DCC method and comparing it with the conventional DCC approach.

It was established multiple states, with each active state allocating the appropriate transmission power and transmission rate according to the current channel conditions.

The authors proposes a mathematical model to evaluate the required power and transmission rate allocation.

The problem is real and the algorithm may provide a good way to solve V2X communications.

The paper is well written and the results are good.

My questions are:

If according algorithm 1, the transmission rate allocation is defined by the TX power, the author are controlling just one variable and creating an association with the second one, which is logical for this problem, but it must be clear the statement that you are controlling two variables.

Although the algorithm demonstrates a mathematical background with several variables it seems the problem could be drastically simplified by checking the congestion and adjusting both TPC and TRC proportionally. Please, comment and demonstrate.

The simulations also need better explanation.

Please, improve all the figures. The text quality is much superior than the graphics.

Author Response

We have attached the response in the attachment.

Reviewer 2 Report

Abstract: Please focus the abstract on your study and your results.
The authors should specify more details regarding the Experiment for the proposed HPR-DCC
The authors should provide more details regarding the analysis of the results.
I suggest a significant rewrite of the introduction. It should provide an overview of the importance of the main contribution of the proposed.

The advantage and disadvantages of the work are suggested to be highlighted in comparison with extant studies or methods.

How to initialize the proposed HPR-DCC?

Some additional experiments are required:
a. - Scalability
b. - Runtime
c. - Memory
d. - Sensitivity analysis

It is necessary to discuss the complexity of the proposed HPR-DCC.
Some syntax errors or improper expressions exist in the manuscript.
More up-to-date studies are suggested to be cited.

Some syntax errors or improper expressions exist in the manuscript.

Author Response

We have attached the response to the attachment.

Please see the atttachment.

Round 2

Reviewer 1 Report

The authors' response document does not contain my questions. The authors must have forgotten to send the answers to my questions, or they may have mistakenly swapped files when submitting this new version. Another possibility is that the system mixed up the order of reviewers. Either way, I don't have the document with my answers, and I haven't identified my inquiries in the final text.

Reviewer 2 Report

Some syntax errors or improper expressions exist in the manuscript.
More up-to-date studies are suggested to be cited.

Some syntax errors or improper expressions exist in the manuscript.
More up-to-date studies are suggested to be cited.
